# Coherent Ising machine based on polarization symmetry breaking in a driven Kerr resonator

Liam Quinn[1,2] ✉, Yiqing Xu [1,2], Julien Fatome[3], Gian-Luca Oppo [4], Stuart G. Murdoch [1,2], Miro Erkintalo [1,2] & Stéphane Coen [1,2]

Time-multiplexed networks of degenerate optical parametric oscillators have demonstrated remarkable success in simulating coupled Ising spins, thus providing a promising route to solving complex combinatorial optimization problems. In these systems, referred to as coherent Ising machines, spins are encoded in the oscillator phases, and measured at the system output using phase-sensitive techniques, making intricate phase stabilization necessary. Here, we introduce an optical Ising machine based on spontaneous polarization symmetry breaking in a coherently driven fiber Kerr nonlinear resonator. In our architecture, the spins are encoded in the polarization state, allowing robust, all-intensity readout with off-the-shelf telecom components. By operating in a newly-discovered regime where nonlinearity and topology lock the system's symmetry, we eliminate drift and bias, enabling uninterrupted Ising trials at optical speeds for over an hour, without manual intervention. This all-fiber platform not only simplifies the hardware but also opens a path to more stable, high-throughput coherent optical optimization devices for applications from finance to drug design and beyond.

Ising machines are analogue computational systems which can efficiently solve complex combinatorial optimization problems, which arise in various fields such as biology, financial modeling, drug discovery, and machine learning[1–5]. Numerous realizations have been demonstrated in recent years, such as those based on trapped ions[6,7], superconducting circuits[8], electronic oscillators[9], spatial light modulation[10], phase-transition nano-oscillators[11], and microresonator solitons[12]. These devices consist of a network of coupled physical elements, each of which can occupy one of two stable states that encode binary spin variables to realize a network of artificial spin states. By judiciously coupling those spins to one another, the entire system can be designed to emulate the Ising model[13]. In particular, measurement of the collective output of the network of artificial spins allows one to infer the minimum energy (ground) state of the Ising Hamiltonian $H = -\sum_{ij} J_{ij}\sigma_i\sigma_j$, where $J_{ij}$ describes the coupling and $\sigma_i = \pm 1$

denotes the spins. Because many important combinatorial optimization problems—that cannot be efficiently solved using classical computers—can be represented as the minimum energy search of the Ising Hamiltonian, Ising machines offer a potentially groundbreaking approach for a diverse range of problems[1,14,15].

Networks of degenerate optical parametric oscillators (DOPOs) represent one of the most successful physical implementations of Ising machines. Known as coherent Ising machines (CIMs), these devices leverage the bistable output phase of each oscillator to realize a network of artificial, binary spin states[13,16–20]. They are particularly appealing owing to their ability to operate at room temperature, provide arbitrary all-to-all connections between spins, enable parallel processing which imparts inherent scalability, and efficiently solve large optimization problems with excellent speed and efficiency[21,22]. Yet, despite their evident successes, today's state-of-the-art CIMs still

[1]Physics Department, The University of Auckland, Auckland, New Zealand. [2]The Dodd-Walls Centre for Photonic and Quantum Technologies, Auckland, New Zealand. [3]Laboratoire Interdisciplinaire Carnot de Bourgogne (ICB), UMR 6303 CNRS, Université Bourgogne Europe, 9 Avenue Alain Savary, Dijon, France. [4]SUPA and Department of Physics, University of Strathclyde, Glasgow, EU, Scotland. ✉e-mail: liam.quinn@auckland.ac.nz

exhibit limitations. In particular, complex phase-stabilization is required to maintain appropriate spin coupling, and to achieve robust read-out of the network state via homodyne detection[17,18,23]. Achieving perfect stabilization of the entire system remains challenging, often necessitating post-selection procedures to reject any of the trials that drift out of phase, thereby limiting the overall speed and efficiency with which the system can find a sufficiently good solution[22,24,25].

Here, we experimentally demonstrate proof-of-concept results of a novel optical Ising machine design, that addresses some of the limitations of existing systems. We circumvent the inherent complexities of representing spin states in optical phase, as used in the CIM architecture, by instead using optical polarization, whereby spins can be discriminated via straightforward intensity measurements. We realize polarization spins by leveraging spontaneous polarization symmetry breaking that manifests itself in coherently driven optical fiber Kerr resonators[26–28]. Moreover, by operating in a recently discovered topological symmetry protected regime[29,30], our artificial spins are intrinsically insulated from unwanted biases, enabling robust Ising operation without the need for post-selection. In demonstrations employing chains of up to 100 spins with all-optical dual neighbor coupling, we achieve continuous repeated measurements of the Ising spin evolution for periods exceeding one h. Compounded by the fact that our novel Ising machine operates at 1550 nm wavelength and makes exclusive use of off-the-shelf telecommunications components, we believe that our implementation offers a promising route towards improved performance, robustness, and stability of CIMs.

## Results

### Polarization spins

We begin by describing the origin of the artificial polarization spins that underpin our concept. Our Ising machine is built around a passive optical Kerr ring resonator formed from a segment of single-mode optical fiber that is closed on itself with a standard fused fiber coupler that is externally, coherently-driven by a single pump laser (see Fig. 1a and Methods). The resonator exhibits two orthogonal polarization modes, corresponding to the principal polarization states that map back to themselves at the end of each round trip. We denote the complex amplitudes of the intracavity electric field along the two polarization modes of the resonator as $E_1$ and $E_2$.

Our system incorporates three polarization controllers (PC1, PC2, and PC3 in Fig. 1a). PC1 is used to align the polarization of the external drive along one of the principal polarization modes of the resonator (here assumed $E_1$). PC2, positioned inside the resonator, acts as a localized birefringent defect. It is configured to introduce a $\pi$ relative phase shift between the resonator modes, so as to operate in the symmetry protected regime described in ref. 29 and briefly recounted below. Finally, PC3 is placed at the output of the resonator and is part of the spin read-out stage.

At sufficiently high pump power, the resonances of the fiber ring are tilted due to the Kerr nonlinearity (see Fig. 1b). The required driving power threshold is comparable to that associated with the occurrence of temporal cavity solitons, as well as other localized and periodic structures of Kerr resonators[26,27,29]. When the external driving field is detuned from the resonance, the intensity of the driven mode $|E_1|^2$ is low and the undriven mode is empty, $|E_2|^2 \approx 0$. However, as the driving frequency is tuned towards the peak of the tilted resonance, parametric four-wave-mixing causes the undriven mode $E_2$ to grow (Fig. 1b, red curve) with two possible phase shifts relative to the driven mode, $\phi_0$ and $\phi_\pi = \phi_0 + \pi$[29]. Furthermore, the $\pi$ phase shift caused by the birefringent defect (induced by PC2) forces a periodic round-trip to round-trip swapping between the two phase states, i.e., $\phi_0 \rightleftharpoons \phi_\pi$ (Fig. 1c).

The bistable phase of $E_2$ is associated with two different polarization states for the intracavity field. These can be resolved in intensity by projecting the resonator output (using PC3 and a polarizing beam-splitter, PBS) onto hybridized modes defined as $E_\pm = (E_1 \pm iE_2)/\sqrt{2}$. In terms of these modes, one polarization state is associated with intensities $|E_+|^2 = I_{max}$ and $|E_-|^2 = I_{min}$ and the other vice versa. Moreover, the periodic swapping of the phase of $E_2$ results in the periodic swapping of the hybridized mode intensities $|E_+|^2 \rightleftharpoons |E_-|^2$ (Fig. 1c, right axis). Because the initial state is selected randomly, from noise, a measurement of the intensity of one of the hybrid modes (say $|E_+|^2$) therefore yields one of the two unbiased sequences: $(I_{max}, I_{min}, I_{max}, I_{min}, ...)$ or $(I_{min}, I_{max}, I_{min}, I_{max}, ...)$, which define our artificial spin states, $+1$ and $-1$ (Fig. 1d). Taking into account the parity of the round-trip index, these can be unequivocally discriminated by a single intensity measurement[30]. Note that the periodic swapping of our artificial spins can be seen as the excitation of Floquet states, which have been recently suggested to offer advantages for escaping local minima of a target Ising Hamiltonian[31].

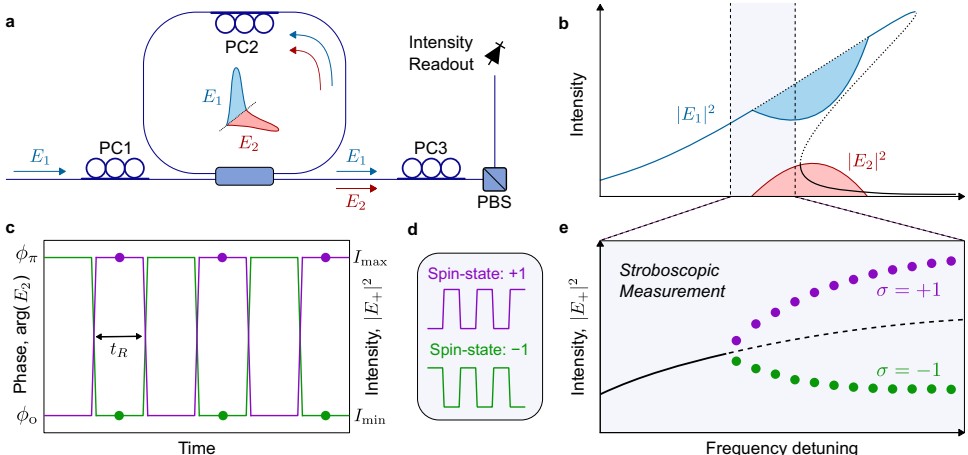

**Fig. 1 | Origin of artificial polarization spins. a** Schematic illustration of the core system components. PC: polarization controller, PBS: polarizing beam-splitter. **b** As the external driving field (along cavity mode $E_1$, blue) is tuned into resonance, the undriven mode $E_2$ is parametrically generated (red) with bistable phases $\phi_0$ and $\phi_\pi = \phi_0 + \pi$ relative to the driving field. Solid (dotted) curves represent stable (unstable) states and the shaded regions highlight depletion (gain) of $E_1$ ($E_2$) due to parametric coupling. **c** Due to the $\pi$ phase shift induced by PC2, the phase of the parametrically-generated field $E_2$ swaps after each round-trip time $t_R$ (left axis), which corresponds to an alternation of the hybridized polarization mode intensities, e.g., $|E_+|^2$ (right axis). Depending on the initial phase of $E_2$, two distinct binary sequences exist (green and purple), which define two artificial polarization spin states, as shown in (**d**). **e** When observed stroboscopically over two round trips, the generation of the phase bistable field $E_2$ corresponds to a spontaneous symmetry breaking (pitchfork) bifurcation in the hybrid mode intensity.

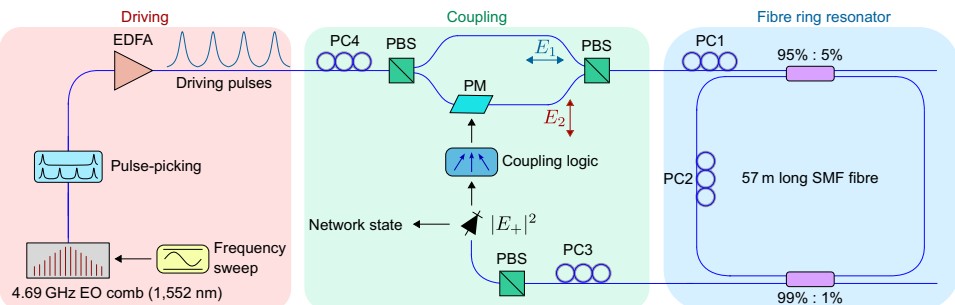

**Fig. 2 | Experimental setup of the polarization coherent Ising machine.** 5 ps pulses derived from an EO comb source synchronously drive a Kerr fiber resonator along the $E_1$ polarization mode, defining time-multiplexed artificial spins (see also Fig. 1). The spins are read out by measuring the hybrid mode intensity $|E_+|^2$ and coupled together through feedback via phase-modulation (PM) of a small driving component along the $E_2$ mode. PC: polarization controller, PBS: polarizing beam-splitter, EDFA: erbium-doped fiber amplifier.

We must emphasize that the localized birefringent defect induced by polarization controller PC2 is crucial to obtain robust, bias free spins. As described in ref. 29, this defect introduces an attractor in the dynamical system that fundamentally eliminates the effect of all asymmetries, even in scenarios involving an imperfect $\pi$ phase shift or a misaligned driving polarization state. This balancing of asymmetries has been experimentally shown to remain robust even when the intended $\pi$ phase defect deviates by more than a cavity linewidth[29,32]. It is also worth noting that, when examined over two round trips, the growth of the undriven mode $E_2$ corresponds to a symmetry-protected polarization spontaneous symmetry breaking (SSB) bifurcation for the hybridized mode intensities $|E_\pm|^2$ (see Fig. 1e), highlighting the suitability of the resultant states to act as spins of an Ising machine[33–35].

## Experimental implementation

For experimental demonstration, we use a 57-m-long resonator constructed from standard MetroCor single-mode optical fiber closed on itself with a 95/5 coupler (see also "Methods"). As shown in Fig. 2, the resonator incorporates a polarization controller (PC2 as described above) and a 99/1 tap-coupler through which the intracavity field is extracted, yielding an overall measured cavity finesse of 42. We coherently drive the resonator with a train of 5 ps pulses derived from a 4.69 GHz repetition-rate electro-optic (EO) comb generator seeded by a narrow-linewidth laser at 1552 nm wavelength. The repetition rate is set such that an integer number of pulses fit into the 273 ns round-trip time $t_R$ of the resonator. Each of the driving pulses will undergo the polarization dynamics described above, thus realizing a time-multiplexed network of artificial spins, with the number of spins $N$ adjustable with a pulse picker placed at the EO comb output.

To couple the artificial spins and define an Ising network corresponding to a particular problem, we use a measurement and feedback approach, where the instantaneous state of each spin inside the cavity −determined from the output intensities along one of the hybrid modes (say $|E_+|^2$)−is used to perturb the driving conditions of other spins. Perturbations are applied by adding a weak driving component along the $E_2$ mode, which is phase-modulated based on the desired coupling. The phase-modulator is placed ahead of PC1 in one arm of a Mach-Zehnder configuration set up between a pair of PBSs (see Fig. 2), with the other arm channeling the orthogonal $E_1$ driving field. An extra polarization controller (PC4) controls the amplitude of the $E_2$ driving component, and thereby the overall coupling strength of the network.

Minimization of the Ising energy of the network is obtained by sweeping (red-shifting) the driving laser carrier frequency across the polarization SSB bifurcation point of the Kerr resonator (see Fig. 1e). In the absence of coupling, this frequency sweep causes the artificial spins to randomly select one of the two available states with equal probability[30]; in the presence of coupling, the spins are attracted to the collective configuration that minimizes the Ising energy (this is shown

explicitly at the end of the Methods). That optimal solution can then be read-out by simply measuring the intracavity intensity along one of the hybrid modes ($|E_\pm|^2$).

For a proof-of-concept demonstration, we consider a one-dimensional spin chain with anti-ferromagnetic coupling, where each spin is coupled to its two nearest neighbors with free boundary conditions (see "Methods" for implementation details). Figure 3 illustrates a typical single run of our Ising machine executed on a 64-spin chain. Figure 3a shows the peak hybridized intensity $|E_+|^2$ of one of the intracavity pulses (blue curve), as the driving laser frequency is slowly swept across the SSB bifurcation point over 1000 round trips, corresponding to 273 µs. The red curve envelope highlights the growing differential between the high and low intensity states as the bifurcation develops, while the inset shows the dynamics over a shorter time frame, where the round-trip to round-trip swapping of the hybridized intensity is clearly visible.

For further insights, Fig. 3b shows the evolution of all 64 spins. We now plot differential intensities $I_D$ calculated for each pulse over consecutive round-trips, with the swapping dynamics unwrapped for clarity, i.e., $I_D = (-1)^m (|E_+^{(m+1)}|^2 - |E_+^{(m)}|^2)$ (where $m$ denotes the round-trip index). The solid black curve highlights the evolution of one particular spin, corresponding to Fig. 3a. Initially, noisy fluctuations about zero differential intensity (dashed line) that are driven by the collective state of the network can be observed, before a clear final spin state emerges. Figure 3b also reveals that the spin distribution stabilizes after approximately 600 round trips, corresponding to ~160 µs. Assigning spin states +1 and −1 based on the sign of the signals plotted in Fig. 3b, we can calculate the corresponding evolution of the Ising energy $H$, which is plotted in Fig. 3c (blue curve). Clearly, the Ising network evolves towards a low energy state. The green curves in Fig. 3c represent the evolution of the maximum and minimum Ising energies obtained from 1500 independent numerical simulations of the experiment (see "Methods"). Finally, the inset shows a pictorial representation of the final network state of the system corresponding to (b). The yellow highlights show the system defects, which account for the final energy of this particular realization being higher than the ground state.

To illustrate the statistical behavior of our Ising machine, Fig. 4a shows the distribution of final Ising energies of 64-spin chains obtained over 1500 experimental trials, compared with corresponding numerical simulations (see "Methods"). We note an excellent agreement between both distributions. In particular, we observe that the distributions are concentrated towards the true ground state, which is reached around 20% of the time (in this model, Ising energies range from −63 to +63 in steps of 2). In Fig. 4b, we measure the energy distributions over 1500 trials under conditions of weaker feedback coupling, again finding excellent agreement between experiments and theory. Figure 4c shows results from repeated measurements of the

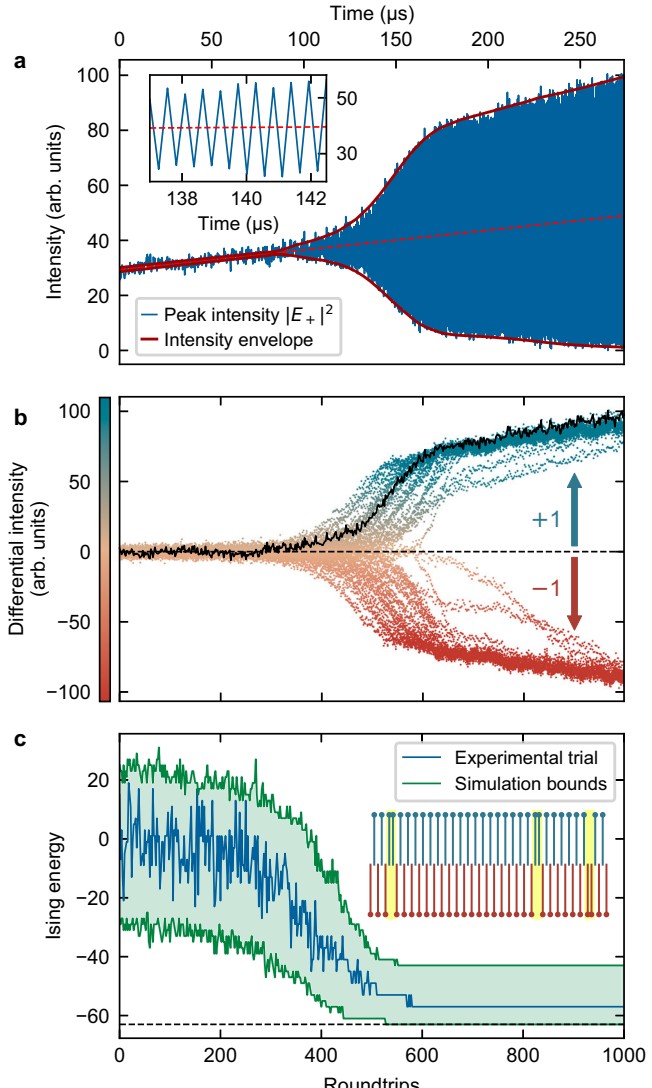

**Fig. 3 | Single run of our Ising machine for a 64-spin chain. a** Peak hybridized intensity $|E_+|^2$ of a single intracavity pulse (blue) as the driving laser frequency is swept over time. The outer red lines highlight the envelope of the evolution, while the zoomed inset reveals the round-trip to round-trip swapping dynamics. **b** Unwrapped differential intensities of all 64 spins over consecutive resonator round trips. The colors map directly to the measured differential intensity, with the black curve highlighting one particular spin. The dashed black line shows the decision threshold for the artificial spins. **c** Corresponding evolution of the Ising energy of the network (blue). The green curves and shading represent the bounds expected from numerical simulations. The inset shows the final network state for the Ising trial shown in (**b**), where yellow highlights mark `defects'.

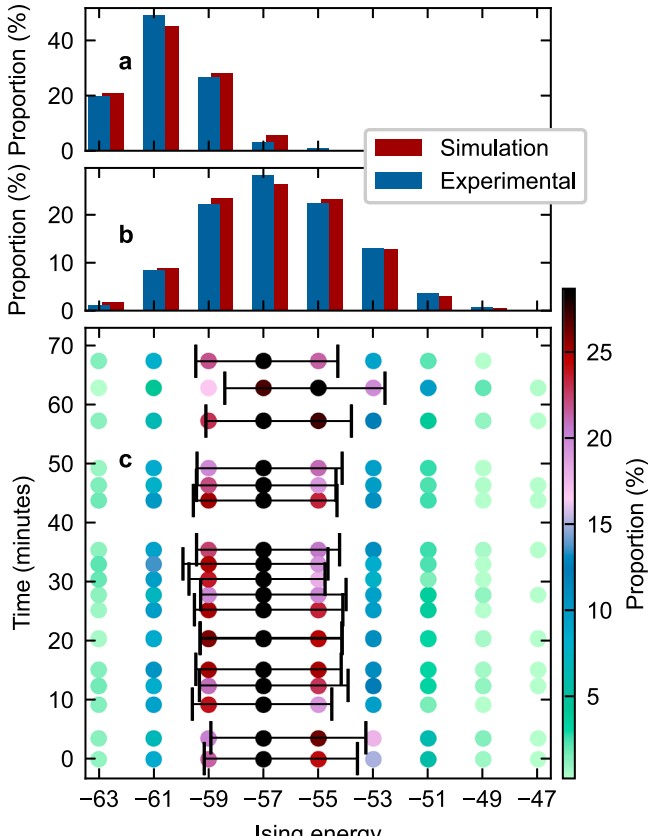

**Fig. 4 | Statistical distribution of final Ising energies obtained for a 64-spin chain. a** Comparison of experimental and simulated distributions obtained over 1500 trials. **b** Energy distribution over 1500 trials under weaker feedback coupling. **c** Repeated experimental measurements of the distribution shown in (**b**) conducted over a time period exceeding one h. No manual adjustments to the setup were required over this time frame and no trials were rejected. The bars in black correspond to one standard deviation.

Ising energy distribution with this weaker coupling, obtained over different batches of 1500 trials recorded over a time period exceeding one h. Remarkably, the energy distribution remains steady over this duration, highlighting the robustness of our Ising machine. Notably, no manual adjustments were made to the setup during the experiment, and every trial was included in the statistics without any rejection (see "Methods").

We now study in more detail our Ising machine's performance and how it scales with the number of spins. The primary metric we consider is the time-to-solution $T_s$, defined as[34]

$$T_s = T_a \left\lceil \frac{\log(0.01)}{\log(1-P)} \right\rceil, \qquad (1)$$

which represents the time required for a certain solution–in our case the ground state–to be found with a 99% probability given the time $T_a$ for a single run of the Ising machine and the probability $P$ for the machine to yield that solution in a single run. For our implementation, $T_a$, also known as the annealing time, is the time taken to sweep the laser frequency through the SSB bifurcation (273 $\mu s$ in the case of Fig. 3; see also "Methods"). Although decreasing the annealing time $T_a$ speeds up the Ising machine's operation, it also reduces the probability $P$ of finding the ground state. Accordingly, one can expect the existence of an optimal annealing time. This can be observed in the data presented in Fig. 5a, where we plot experimentally obtained time-to-solutions $T_s$ as a function of annealing time $T_a$ for different spin chain lengths (see "Methods"). Our experimental data reveal that the optimal annealing time, marked by the black boxes where the minima in the data are located, scales with the system size, with larger systems requiring longer individual run times to minimize the time-to-solution.

Finally, Fig. 5b and c illustrate how the time-to-solution of our Ising machine scales with the number of spins $N$. It has been suggested that a major advantage of DOPO-based coherent Ising machines is that they can outperform other well-studied optimization platforms, such as classical neural networks, with a performance advantage that becomes increasingly substantial for large problem sizes[36]. While extensive studies of the dependence of the time-to-solution on problem size remain relatively scarce[24,37], experiments shown in Fig. 5b, c suggest that our polarization-based Ising machine is consistent with a scaling on the order of $\exp(\sqrt{N})$.

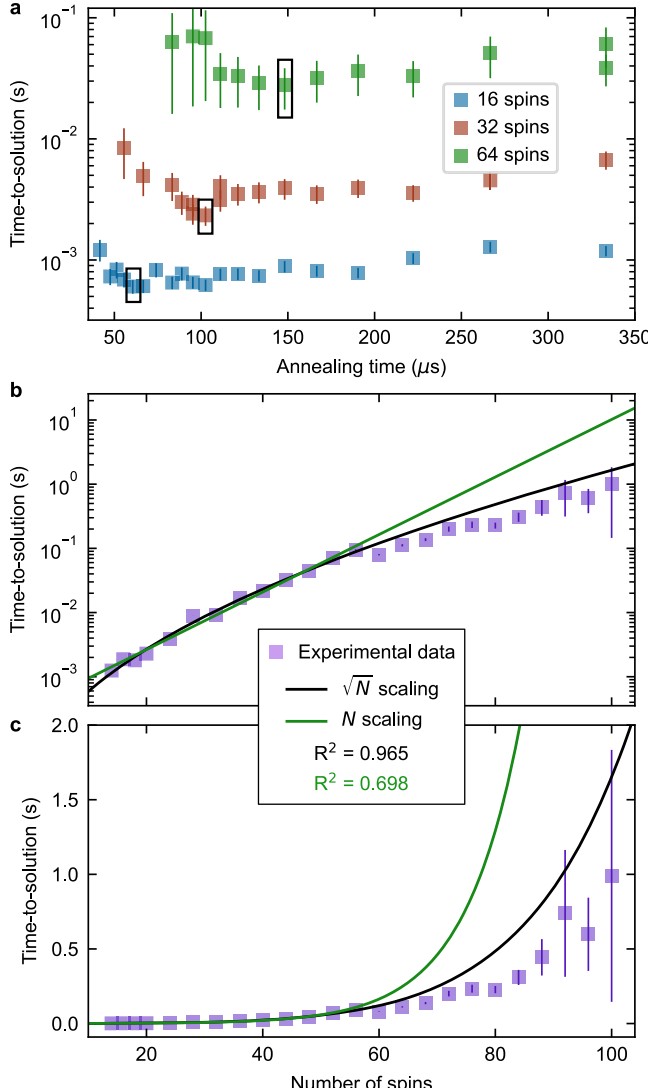

**Fig. 5 | Time-to-solution of the polarization Ising machine. a** Experimental time-to-solution $T_s$ versus annealing time $T_a$ for different spin chain lengths. The black boxes highlight optimal annealing times. **b**, **c** show measured time-to-solution (filled squares) as a function of the number of spins in logarithmic and linear scale, respectively. The data for spin numbers up to 56 are fitted to $e^{\sqrt{N}}$ (black curve) and $e^N$ (green curve) models. The corresponding $R^2$ parameters are evaluated based on the entire dataset. Vertical bars indicate one standard deviation.

The data shown in Fig. 5b, c were obtained by considering chains of $N = 10$ to $N = 100$ spins, with the annealing time set for each case to the optimal value estimated from measured data (Fig. 5a). We then fitted the measured time-to-solutions obtained for 10 to 56 spins to two different scaling models, $\exp(\sqrt{N})$ (black curve) and $\exp(N)$ (green curve). Comparing those fits to the entire dataset, as shown in Fig. 5b and c, provides strong evidence of performance that indeed aligns with the $\exp(\sqrt{N})$ scaling. The $R^2$ parameters for the fits, evaluated for the entire dataset and displayed on the plot, further reinforce these findings. Notably, for trials extending up to 100 spins, the $\exp(\sqrt{N})$ scaling consistently serves as an upper bound for the data, underscoring its applicability to our system (see also "Methods"F). This provides evidence that our Ising machine may scale well for larger and more useful problems, and similar tests will be applied to networks of greater connectivity and more complex topology in future studies.

## Discussion

Our work demonstrates that spontaneous polarization symmetry breaking in externally-driven Kerr resonators enables a novel form of optical coherent Ising machine, in which spins can be discriminated with straightforward intensity measurements. Our scheme critically leverages the recent discovery that a localized birefringent defect can protect the symmetry of the system[29], thus ensuring stable and bias-free polarization spins.

We have presented proof-of-concept experimental results showcasing continuous operation of the machine with up to 100 coupled spins for over an hour, with no need for any post-selection or manual adjustments. The observed scaling behavior, consistent with an $\exp(\sqrt{N})$ dependence, underscores the platform's potential for solving larger, more complex problems. Although stabilizing long fiber cavities is inherently challenging, our method avoids the need for additional phase stabilization in detection and feedback, offering a simple and robust paradigm.

We note that our system can still be optimized, in particular by adjusting the finesse of the resonator. A lower finesse, which translates into a shorter resonator photon lifetime, would speed up the overall dynamics, enabling lower annealing time for the same resonator length. Experimentally, this can be trivially achieved by introducing extra losses at the expense of a higher driving power. However, the number of round trips available to apply perturbations would also be reduced. This suggests a trade-off between speed, driving power (finesse), and solution quality similar to the findings presented in Fig. 5a.

Because the focus of our work has been on proof-of-concept demonstration of the overall scheme, the coupling topology has been limited to a comparatively simple one-dimensional spin chain, yet we emphasize that all-to-all coupling can be readily achieved in our scheme by using established methods based on field-programmable gate arrays (FPGAs)[17,18,22]. We believe that the full telecommunications compatibility of our setup, combined with the fundamental symmetry protection and the ability to resolve the spin states using intensity-only measurements, positions our scheme as a highly promising avenue to solve complex combinatorial optimization problems with unprecedented robustness and stability. We envisage this stability being especially important as investigations towards implementing optical coupling become more feasible[38,39].

## Methods

### Resonator design

Our ring resonator is built around 57 m of MetroCor single-mode fiber. The fiber has a Kerr nonlinearity coefficient $\gamma \approx 2.5\,W^{-1}\,km^{-1}$ and it exhibits normal group-velocity dispersion at the 1552 nm driving wavelength. Note that normal dispersion is important to suppress modulation instabilities that may otherwise impact the dynamics. The ring includes a 95/5 fiber coupler for injection of the driving field and a 99/1 tap coupler for extraction and monitoring of the intracavity field. Overall, the resonator has a total round-trip time $t_R = 273$ ns, corresponding to a cavity free-spectral range FSR $= 1/t_R = 3.666$ MHz, and a cavity finesse of about $\mathcal{F} = 42$. This corresponds to 15% power loss per round trip and to a photon lifetime of 1.8 μs, which characterizes the relaxation time of the system. The detuning scan must remain slower than this lifetime to ensure sufficient dissipation and allow the system to relax back to the symmetric state between trials.

### EO comb generator

The resonator is coherently driven with a train of 5-ps-long pulses generated from an electro-optic (EO) frequency comb seeded with a 1 kHz-linewidth continuous-wave (cw) laser. The cw beam is first passed through a phase modulator, followed by an intensity modulator[40].

Both modulators are driven in phase by the same RF clock synthesizer[40]. The resultant comb is then spectrally broadened through 2.2 km of dispersion-compensating fiber, and subsequently undergoes nonlinear (soliton) compression through a 1 km-long segment of SMF-28 fiber. Finally, a nonlinear-amplifying loop mirror eliminates the residual low-intensity background existing between the pulses. The RF clock of the EO comb determines the repetition rate of the generated pulses. In our case, it is set at 4.6928 GHz, corresponding to 1280 × FSR, thus ensuring synchronized driving of the resonator. The desired number of spins $N$ (i.e., pulses per round-trip) is selected with a pulse picker implemented with an electro-optic modulator driven by a pulse-pattern generator. The spacing between the intracavity pulses is set to be 0.85 ns. This is wide enough to avoid any potential tail interactions between adjacent pulses, guaranteeing the independence of each spin. Our system operates reliably over a range of driving powers, from 0.4 to 1 W peak power. When we vary the number of spins, the EDFA is adjusted to keep the peak power of the driving pulses within that range.

## Laser frequency sweep across the SSB bifurcation

To maintain the fiber resonator near the polarization SSB bifurcation point while we sweep the driving laser frequency, we use the technique of ref. [41] to actively stabilize the detuning between the driving laser and a cavity resonance. Specifically, a low-power cw signal derived from the driving laser is frequency-shifted via an acousto-optic modulator and launched into the resonator in the counter-propagating direction relative to the primary driving pulses associated with the Ising spins. The intracavity power level of this signal is then locked to a setpoint using a PID controller that actuates the driving laser frequency through a piezoelectric transducer in the laser head, thus actively stabilizing the frequency detuning.

The driving laser frequency can then be periodically swept around the setpoint to anneal the spins and run the Ising machine by superimposing a sinusoidal voltage on top of the PID feedback signal. The annealing time $T_a$ referred to in the text then corresponds to half the sinusoidal period. In this way, many Ising runs can be performed consecutively, without any resetting protocol.

## Implementation of the 1D Ising spin chain

The 1D Ising spin chain coupling topology is implemented by placing a 50/50 beamsplitter behind the PBS at the cavity output. The two output ports of the beamsplitter are each followed by an optical delay line, before photodetection. The electrical signals from the two photodetectors are then combined before being applied directly to the phase-modulator acting on the $E_2$ driving component. The optical delays are adjusted so that each spin influences the driving pulse corresponding to the spin immediately up and down the chain, respectively, at the next round-trip. As our $N$ spins never fill up the entire resonator, the two spins at the edge of the chain are only coupled to one neighbor each, which effectively corresponds to free boundary conditions.

## Coupling calibration and stabilization

The two arms of the Mach-Zehnder interferometer used to phase-modulate the $E_2$ driving component are typically affected by a slow long-term relative phase drift (over several seconds). Such drift effectively affects the overall polarization state of the driving beam, which we monitor with a commercial polarimeter before the driving beam is injected into the resonator. To achieve stable operation, an error signal derived from a combination of multiple Stokes parameters read out from the polarimeter is fed into a PID controller acting on a fiber strecher placed in one arm of the Mach-Zehnder interferometer. The optimal setpoint of the PID is obtained by a calibration routine, whereby we step the fiber stretcher while running the Ising machine continuously. In this way, we can determine the driving beam Stokes

parameters yielding the minimal Ising energy (with averages taken over 300 runs). Once the calibration is performed, stable operation can be maintained for periods exceeding one h.

## Determination of time-to-solutions, $T_s$

The time-to-solutions plotted in Fig. 5 are determined for each point out of 1500–4500 runs of the Ising machine. From these batches of results, we extract each time the probability $P$ of reaching the ground state, which is then introduced into Eq. (1).

## Numerical model

The numerical simulation results presented in Figs. 3c and 4a were obtained with a simplified model where the individual spins are represented as cw fields. Minimal differences were observed when considering the full fine temporal structure of the spins, and this was neglected for computational efficiency. Specifically, we iterate, for each spin, the following Ikeda map, which corresponds to the boundary conditions of our fiber resonator[42],

$$E_{1,i}^{(m+1)}(0) = e^{-\alpha} E_{1,i}^{(m)}(L) e^{-i\delta_0} + \sqrt{\theta}\, E_{in} \cos\chi, \qquad (2)$$

$$
\begin{aligned}
E_{2,i}^{(m+1)}(0) = &\, e^{-\alpha} E_{2,i}^{(m)}(L) e^{-i(\delta_0 - \pi)} \\
&+ \sqrt{\theta}\, E_{in} (\sin\chi)\, e^{i\phi_i^{(m+1)}}.
\end{aligned}
\qquad (3)
$$

Here, $E_{1,i}^{(m)}(z)$ and $E_{2,i}^{(m)}(z)$ represent the electric fields of the two polarization modes of spin $\sigma_i$ at the $m$-th round trip, $L = 57$ m is the resonator length, $\theta = 0.05$ is the power transmission coefficient of the input coupler, with all other losses lumped into $\alpha = \pi/\mathcal{F} \approx 0.075$, while $\delta_0$ is the round-trip phase detuning of the $E_1$ mode (swept from 0 to $0.8\,\alpha$), with the equation for the second mode also including the $\pi$ phase shift defect for topological symmetry protection[29]. $E_{in}$ is the amplitude of the driving field, with $P_{in} = |E_{in}|^2$ the total (peak) driving power, and $\chi$ represents the effective driving polarization ellipticity, reflecting the setting of PC4. These were set to match with experimental observations. Specifically we used $P_{in} = 0.4$ W and $\chi = 0.1$ rad for the simulation results of Fig. 3c while $P_{in} = 0.7$ W and $\chi = 0.1$ rad and 0.05 rad, respectively, in Fig. 4a, b. Finally, $\phi_i^{(m+1)}$ accounts for the $E_2$ driving phase modulation through which we implement the coupling between the spins and is given by

$$\phi_i^{(m+1)} = g \sum_j J_{ij} |E_{+,j}^{(m)}|^2. \qquad (4)$$

Here $J_{ij}$ describes the coupling topology while $g$ relates to how strong we amplify the electronic signal driving the phase modulator and which—together with $\chi$—affects the overall coupling strength of the spin network. The value of $g$ in numerical simulations is initially estimated based on experimental measurements, and then further fine-tuned to match the experimental results. We used $g = 0.4$ for 64 spins (as in Figs. 3 and 4), with a scaling based on the number of spins and coupling matrix as described in ref. [24]. Finally, environmental noise is represented by adding weak uncorrelated white noise with random phase and amplitude to the driving field $E_{in}$, thereby seeding the onset of symmetry breaking.

Propagation along the resonator round-trip from $z = 0$ to $L$ is described by coupled nonlinear wave equations of the form

$$\frac{\partial E_{1,2}(z)}{\partial z} = i\gamma(|E_{1,2}|^2 + B|E_{2,1}|^2)E_{1,2} + i\gamma C E_{1,2}^* E_{2,1}^2. \qquad (5)$$

Numerical integration is performed with a fourth-order Runge-Kutta method, and assuming modes of linear polarization states (corresponding to $B = 2/3$ and $C = 1/3$)[43,44], with $\gamma = 2.5\,\mathrm{W}^{-1}\,\mathrm{km}^{-1}$ the nonlinearity coefficient of the fiber.

We close this Section by highlighting how the mean-field model that was dervied in the Methods of[29] to describe topological symmetry protection can be generalized to include the additional coupling-induced phase shift $\phi_i^{(m+1)}$ defined above. Over two round trips, we find that the evolution of the $E_2$ mode amplitude picks up an additional driving term of the form

$$\sqrt{\theta}\, E_{in}(\sin\chi)\left[ e^{ig\sum_j J_{ij}|E_{+,j}^{(m+1)}|^2} - e^{ig\sum_j J_{ij}|E_{+,j}^{(m)}|^2} \right]. \quad (6)$$

Under the assumption of small phase perturbation, this becomes

$$ig\sqrt{\theta}\, E_{in}(\sin\chi)\left( \sum_j J_{ij}\left[ |E_{+,j}^{(m+1)}|^2 - |E_{+,j}^{(m)}|^2 \right] \right). \quad (7)$$

The value in bracket, $|E_{+,j}^{(m+1)}|^2 - |E_{+,j}^{(m)}|^2$, matches with our definition of the spins $\sigma_j$ based on differential intensity $I_D$. Hence, the driving perturbation on each spin $\sigma_i$ takes the form of $\sum_j J_{ij}\sigma_j$ and maps to the Ising Hamiltonian[13]. This demonstrates that the intended Ising Hamiltonian is correctly implemented even in the presence of coupling, subject only to limitations imposed by amplitude heterogeneity, which is a constraint shared with existing CIM architectures[37].

## Data availability
The data that support the plots within this paper and other findings of this study are available from the corresponding author upon request.

## Code availability
The code that supports the plots within this paper and other findings of this study are available from the corresponding author upon request.

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

## Acknowledgements

We acknowledge the financial support provided by The Royal Society of New Zealand in the form of Marsden Funding (18-UOA-310 and 23-UOA-053). Additional financial contributions were kindly provided by CNRS through the IRP Wall-IN project and the Conseil Régional de Bourgogne Franche-Comté.

## Author contributions

L.Q. performed all of the experiments with the assistance of Y.X. Numerical simulations were completed by L.Q. with the help of S.C. and M.E. The first draft was written by L.Q. with subsequent editing and review completed by M.E. and S.C. The theory and concept were developed by G.O., J.F., M.E. and S.C. Additional support and supervision were provided by S.G.M. The overall project was supervised by M.E and S.C.

## Competing interests

The authors declare no conflicts of interest.
