## [Peer Review File · Nature Communications]

Coherent Ising Machine Based on Polarization Symmetry Breaking in a Driven Kerr Resonator

Corresponding Author: Mr Liam Quinn

Version 0:

Reviewer comments:

Reviewer #1

(Remarks to the Author)

I reviewed the revised manuscript by Quinn et al., "Coherent Ising Machine Based on Polarization Symmetry Breaking in a Driven Kerr Resonator."

In my original report, I asked the authors to (i) quantify the impact of a non-perfect π defect, (ii) discuss the threshold for the onset of the polarization symmetry-breaking regime relative to other Kerr-driven phenomena, and (iii) expand their outlook on scaling and possible all-optical coupling architectures. All these points have been addressed in the revised manuscript. The tolerance to phase-defect imperfections is now explicitly discussed and supported by references [29, 32]; the power threshold is compared with that of modulation-instability and soliton formation; and the Discussion section now includes a perspective on future all-optical coupling schemes.

I also find Reviewer #3's criticism valuable and constructive. The authors have undertaken a consistent and careful revision in response, and I find that the new version substantially improves the manuscript. In particular, the updated statistical analysis (Fig. 4) and the demonstration of continuous one-hour operation without post-selection highlight a clear enhancement and a genuine focus on system robustness.

The work's novelty lies in the realization of a symmetry-protected, polarization-based Ising machine, introducing a fundamentally new spin-encoding mechanism distinct from previous phase-based CIMs. The added discussion on π -defect tolerance and threshold comparison further consolidates the conceptual foundation.

While the overall computational performance and benchmarking remain below the state of the art, such optimization is primarily a technical goal. The key contribution of this paper is the demonstration of a viable and conceptually new mechanism for topologically protected spin encoding. The manuscript now provides a well-validated proof-of-concept of a symmetry-protected optical computing paradigm, which falls well within Nature Communications' remit to publish technically sound and conceptually significant advances.

Reviewer #4

(Remarks to the Author)

The manuscript is about implementing an Ising machine based on polarization symmetry breaking in a fiber cavity, and demonstrating the ability of this system to solve optimization problems by minimizing the Ising energy.

The paper has already been reviewed (for Nature Photonics) by two reviewers, one of which is rather positive while the other reviewer has some questions about novelty and about the appropriateness of the benchmark problems used in the paper.

The topic of the paper, being a combination of photonics, non-linear dynamics and computing, certainly fits within the scope of the journal Nature Communications. The paper is well-written and clearly structured and the response to the reviewers'

questions address most of their concerns. I believe the paper to be of interest to the research communities working on Ising machines and those working in the field of non-linear optics. The main innovation of the paper for me is that the suggested approach leads to a much-enhanced stability, even in the case of using non-ideal components and/or when drift is present. Moreover, using topology to avoid asymmetries can be an interesting scheme for other Ising machine implementations.

Although I do agree with most of the claims of the paper, I think some more work is needed to further substantiate the conclusions. My main concern is related to possible unintentional effects influencing the proper operation of the Ising machine. The authors clearly motivate that the localized defect results in bias free spins in the case that there is no coupling between the spins, which is an important effect. But can they also prove/show that this leads to the implementation of the intended Ising Hamiltonian when there is coupling? Related to this issue is the remark made by the authors that the balancing of the asymmetries remains robust even when there is a deviation of the π phase defect. Again, without coupling this statement seems to be justified, but can the authors further elaborate on this issue when operating the coherent Ising machine (i.e. when there is coupling)? Would it then be possible that the Ising machine actually oscillates between 2 ground-states that are both (slightly) different from the ground-state of the initial problem at hand?

Some other, minor comments are

1. Related to the comment on the threshold and power dissipation, it would be illustrative to add some information about injected power and power dissipation in the methods section
2. To make results reproducible, please specify all parameter values used for the simulations in Figs 3, 4 and 5. There is an index missing in $\phi^{(m+1)}$ in Eq (3)
3. How do you match the value of parameter g (the coupling strength) in the model with the experimental parameters to match the experimental and the numerical distributions shown in Fig 4 (a) and (b)? Is this correspondence derived from other, independent measurements or are the parameters matched based in these distributions themselves?
4. When discussing the use of an FPGA for all-to-all coupling, the coupling will typically be much slower than the resonator's round trip time. Do you foresee extra difficulties with matching the feedback signal to the correct round-trip? Will this require a precise clocking of all signals?
5. Related to the comment on the number of round trips needed to reach a solution, it is true that the system of the paper is fast as the round-trip time is short, leading to an annealing time of only 60 ns. Still, the fact that a (somewhat) larger number of iterations is needed is interesting in itself, as it seems to suggest that the inherent time-scale at which the Ising machine fluctuates is somewhat slower (as expressed in number of iterations). Can the authors explain these differences and pinpoint which system parameters influence this inherent time-scale?

As a final comment, I think the paper will become suitable for publication after the authors address the main concern detailed above.

Version 1:

Reviewer comments:

Reviewer #4

(Remarks to the Author)

The authors have carefully addressed in their response letter the various questions and comments I raised and have made appropriate changes to the manuscript. In particular, my main comment regarding the implementation of the expected Ising Hamiltonian (in the presence of coupling) has been clarified, and this discussion further supports the claims of the paper. The detailed analysis of the dynamics and their dependence on the finesse is also valuable. I find Figure A in the response letter particularly illustrative. Although I acknowledge that it may not fit within the main manuscript, I encourage the authors to include this figure and its description as supplementary material, but I leave it up to the authors to decide on this point.

With all my comments being taken care of, I recommend the paper to be published in Nature Communications.

Summary of Changes made:

Reviewer 1 considered our original submission as suitable for publication in Nature Photonics, and provided a very positive Review with only minor comments. We have been able to address all these points in our revised submission. In particular, we have included additional references to related and important work, which in particular clearly address how protected symmetry-breaking responds to non-ideal conditions, i.e. when the phase defect differs from an exact value of π .

Reviewer 3 was more critical and raised three main points about novelty and contribution, phase stability and architecture, and performance and benchmarking. All of these are discussed in more detail below. In terms of changes, we are now better insisting that our work should be seen as a proof-of-principle demonstration of a novel promising Ising machine platform (for which full benchmarking is out of scope and left for future work). In particular, we have clarified the novel aspects of our approach: how it exploits a new regime of polarization symmetry-breaking and constitutes the first experimental demonstration of a period-2 Floquet Ising solver, and how our platform removes two layers of feedback, fundamentally reducing stabilization requirements. Additionally, we have clarified why our ground state success rates appeared (artificially) so low and have included other results demonstrating an optimized 20% success rate for 64 spins. Finally, we now explain that due to our shorter cavity length, our effective annealing time is faster than prior systems despite the larger number of round trips.

All reviewer comments have been included verbatim, and we attempt to address these comments point by point to the best of our ability.

Reviewer 1

The paper by Quinn et al., "Coherent Ising Machine Based on Polarization Symmetry Breaking in a Driven Kerr Resonator," presents a coherent Ising machine built on the authors' recent discovery regarding symmetry breaking in the polarization state. Specifically, the authors use topological nonlinear protection of symmetry breaking to create artificial spin states. Such a state is achieved via a localized π defect in the fiber ring, which forces optical parametric Kerr generation on the orthogonal linear polarization. The spins are encoded in the circularly polarized state of the system, which acquires randomness in the process. With this approach, the authors experimentally demonstrate operation with up to 100 spins in a time-multiplexed network, using a 4.69 GHz electro-optic comb as the pulse source. The system, which operates with up to 100 spins and remains continuously stable over long durations, shows that its time-to-solution scales as $\exp(\sqrt{N})$, indicating its potential for efficiently solving complex combinatorial optimization problems. I find the paper well written, based on solid physics and of strong impact. To further strengthen the paper, and to provide a perspective for other types of technologies (e.g., integrated) based on this approach, I would suggest:

Comment 1

Can the authors please explicitly show the reader the limitation of a non-perfect π defect? From my understanding, an imperfection in the defect would mainly affect the visibility of the oscillation in the circular polarization (e.g., the max and min intensity in Fig. 3a). Can this be quantified? I stress that, if I have understood this point correctly, the fact that the defect affects only the visibility (and not the existence) of the spin state is a fundamental strength of the paper. A π defect has been explored in integrated configurations (Wang, J., Valligatla, S., Yin, Y. et al., Nat. Photon. 17, 120–125, 2023), which is generally difficult to realize perfectly, and I think an explicit definition of the phase error's impact could enhance the paper's relevance for integration.

The Reviewer makes insightful comments about the details underpinning the exact π -phase shift on the performance of the system. Indeed, deviations from an exact π -phase shift only affect the visibility of the spin-states, and does not impose any bias. In fact, this is something that we have explicitly studied in a previous work [29]. Specifically, in Fig. S4 in the Supplementary Information of that article, we perform randomness tests for uncoupled spins, and show that, upon symmetry breaking, the spin selection stays perfectly random even for deviations of the π phase defect corresponding to several resonance widths. We also comment that the limit is imposed by our eventual inability to distinguish between the spin states (i.e. loss of visibility). Because these results are already published, we have simply provided appropriate references ([29, 32]) where this phenomenon is discussed in depth. We now also explicitly state that the system is robust under non-idealized π -phase defects.

Comment 2

In the same vein, could the authors please discuss the threshold of formation of this phenomenon, comparing it, for instance, to the threshold of generation for other well-known phenomena in Kerr-driven resonators (e.g., modulation instability patterns or temporal cavity solitons)? This would help those in the integrated photonics community draw a clearer parallel for building such an Ising machine on an integrated platform.

We presume that the Reviewer is referring here to the threshold of spontaneous symmetry breaking (the π phase defect and the associated alternation of the polarization states leave that threshold unchanged as is discussed in [29]). This is something that has been well studied before (e.g. in [26] and in other references we cite in [29]). In fact, spontaneous symmetry breaking occurs in a Kerr resonator on the upper branch of

the bistable response, and in that sense the driving powers involved are very similar to that required for the generation of cavity solitons or modulation instability (in the anomalous dispersion regime). We have now added a specific comment in the “Polarization spins” section to address this point.

Comment 3

Scaling of Ising machines beyond a single chain is always a challenge. The authors mention all-to-all coupling with standard FPGA-based methods, but given their nonlinear implementation, do they think it might be feasible to use other all-optical approaches? There are recent works (e.g., Gray, R. M. et al., arXiv:2405.17355v1) that discuss different methodologies. I think that a detailed discussion is beyond this paper’s scope, however the authors could highlight how all-optical approaches may ultimately be key to moving beyond FPGA-based controls, which are the real bottleneck of time-multiplexed Ising machines.

This is a good point, which we now address in our Discussion at the end of our manuscript. We now provide an outlook that the increased stability offered by our platform may be of great use for the eventual transition to all-optical or hybrid optical-electronic feedback schemes. This is supported by additional references that allude to the possibility of realizing such all-optical couplings.

Final Comment:

I have no further suggestions, and I think the paper is suitable for Nature Photonics.

Reviewer 3

The manuscript under consideration introduces an optical coherent Ising machine that leverages spontaneous polarization symmetry breaking in a Kerr-nonlinear fiber resonator to encode binary spin states via polarization rather than phase. The authors demonstrate a time-multiplexed network of up to one hundred spins, maintained in a topologically protected symmetry regime to suppress bias and drift, and report empirical $\exp(\sqrt{N})$ scaling in time-to-solution. The entirely fiber-based, telecom-compatible implementation and ability to read out spin configurations through simple intensity measurements are, at first glance, very appealing features for photonics-enabled optimization hardware. However, despite these promising aspects, the work does not meet the high standard of novelty, broad impact, and significance expected by Nature Photonics.

Comment 1:

While the manuscript is written well and the experimental work appears technically sound and carefully executed, the overall contribution remains just incremental. It ultimately presents what is effectively one more Ising machine. Indeed, any bistable system can in principle be used for the basis of an Ising machine. Therefore, to be published in a top-tier journal, the system must be significantly novel, clearly demonstrate an advantage over existing implementations, and exhibit competitive performance compare to the state-of-the-art.

While the reviewer makes positive remarks on the writing and execution of the experimental work, they state that the system is “effectively just one more Ising machine”. We believe that this comment is a bit on the harsh side but we acknowledge that we may have not stressed well enough the true novelty of our contribution. We have now updated the text of our resubmission to better emphasize the novel aspects of our work.

First of all, we would like to point out a recent preprint by Cassella et al., “Floquet-Based Ising Machines Escape Local Minima in QUBO Problems”, Research Square, June 16, 2025 (doi:10.21203/rs.3.rs-6870975/v1). These authors suggest theoretically that periodic time-variation (which is what “Floquet” refers to) could be a fundamentally improved method for escaping local minima of combinatorial problems. In that context, our current submission can be seen as the first experimental demonstration of a (period-2) Floquet Ising solver, and could therefore be highly relevant in the near future. We have now included that reference (now [31]).

We must also stress that our system also presents the important advantage of defining and measuring spins in intensity, which is not an incremental advantage. In particular, it provides much enhanced stability (see also our reply to Comment 3 below) and avoids having to discard massive amount of data.

In summary, our work includes several important novel aspects and constitutes a powerful new platform for combinatorial problem solving. These points have been incorporated better into our revised submission throughout the text.

Comment 2:

The central innovation—using polarization bistability in a Kerr resonator—draws from the established concept of symmetry breaking in nonlinear optics. Furthermore, the experimental setup was already introduced in two previous publications by the same group (references 28 and 29). It closely resembles an existing coherent Ising machine architecture (e.g. Takesue), which consists of a fiber loop, a nonlinear optical element, and time-multiplexed Ising spins.

Regarding the implication that polarization stability is an old, well-known phenomenon, we emphasize that the alternating regime of symmetry-breaking that we are exploiting (induced by the π phase defect) is quite novel, and has only been discovered and understood in the last year or two. This modal alternation goes beyond

conventional bistability, representing a genuinely novel regime of operation for Kerr resonators. Moreover, we do not believe that the use of a fiber-loop architecture alone should imply that our approach is merely an iteration of existing coherent Ising machines (CIMs). Our system utilizes a *passive* Kerr resonator, with *polarization* defining spin-states. It requires no active control for read-out, and perturbations are applied on the primary driving field. By contrast, the CIM system features an active nonlinear gain element [typically periodically poled lithium niobate (PPLN)] that supports optical parametric oscillations, with Ising spins encoded through bistable phase states, and perturbations applied using additional feedback pulses injected into the cavity.

Comment 3:

The authors claim superior phase stability comparing to previous implementations. While I can agree that phase stabilization occurs at many different point in case of coherent Ising machine, this comparison is ultimately not fair, because traditional coherent Ising machines usually address problem sizes 1000 times larger, requiring longer fiber cavity and inherently more difficult stabilization. Moreover, the demonstrated coupling architecture is limited to a one-dimensional chain with nearest-neighbor interactions. More complex graph topologies—such as all-to-all coupling or embedding of nontrivial graphs—remain not explored, which limits demonstration of the platform’s versatility.

The reviewer is correct in that stabilizing a longer fiber cavity is indeed harder, which will also be an experimental challenge for us in the future as we progress our investigations. However, we must emphasize that our scheme fundamentally removes not one, but *two* layers of feedback (out of three for other CIMs). With our spins defined in intensity, phase stabilization protocols for both homodyne detection, as well as feedback pulse injection, are simply not needed at all, which remains true irrespective of the cavity size.

Here we also must insist that our work should be seen as a proof of concept of a novel Ising machine platform. In that sense, it is unreasonable to expect that we immediately tackle all-to-all-connected large combinatorial problems; we will study both larger and more complex problems in future reports.

Comment 4:

From an application and performance viewpoint, the manuscript falls short by not benchmarking the proposed system with existing coherent Ising platforms or classical algorithm on existing benchmark set and on real-world optimization problem. Without such comparative analysis, practical merit of polarization-encoded CIMs over degenerate OPO systems or electronic annealer is not clear. By focusing only on simple nearest-neighbor chain, authors do not show their system can reach to the state-of-the-art performance in either solution quality or time to solution. Surprisingly, Figure 3 shows 600 round trips are needed to solve nearest-neighbor 64-spin chain—although literature usually shows convergence in fewer than 50 iterations for same lattice structures. Also, Figure 4 shows success rate only 1.5% for reaching the ground state, which is much lower than reported rates for more large and complicated problems in other works. This again proves the need of testing large and hard benchmark problems.

We begin by thanking the Reviewer for pointing out the issue with what appears to be a low success rate of 1.5 % of reaching the true ground state. In the original Fig. 4, we had biased the system on purpose — without mentioning it explicitly — to widen the energy distribution (which lowers the ground state success rate), with the aim to better highlight the repeatability and robustness of our system. When properly optimized the ground state success rate reaches 20 %, which is now shown in a new panel, Fig 4a. We now clearly discuss this point in our revised submission.

Regarding benchmarking, we must insist again that our work should be seen as a proof-of-concept of a novel Ising machine configuration. We therefore focus on the platform description, as well as on the physics

and dynamics of the system, while extensive benchmarking is outside the scope of our paper.

Finally, we would like to argue that the comparison with the number of round trips needed for the system to converge is a poor one, as it does not take into account the fact that our resonator has a significantly shorter round trip time. Consider e.g. the work of McMahon et al (2016), “A Fully Programmable 100-Spin Coherent Ising Machine with All-to-All Connections.” *Science* 354 (6312): 614–17 (doi:10.1126/science.aah5178). In this work, the authors study (quoting)

“... $N = 16$ cubic graphs, of which there are 4060. We were able to find ground states with probability greater than 20% for every single instance (Fig. 2G). In this experiment, and all that follow, every run was set to proceed for $N_{rt} = 300$ round trips, a computation time of $T_{comp} = 480 \mu\text{s}$.”

In our work, for the same number of spins ($N = 16$), Fig. 5 demonstrates an annealing time of $60 \mu\text{s}$, approximately eight times faster than the work referenced above, which directly contradicts the comments made by the Reviewer. Also, the fact that we do not have to discard any data through post-selection gives us a further speed advantage. Our system is therefore competitive.

Comment 5:

Finally, proposed $\exp(\sqrt{N})$ scaling is only demonstrated for a toy problem and may not hold when applied to more difficult or larger problems. Because the analysis is limited to system with less than 100 spins, scaling does not consider practical overheads which will surely become big at larger scales.

The Reviewer is correct to mention that the scaling we have demonstrated only strictly applies to the simple nearest-neighbor chain. We have addressed this by clarifying in our Discussion that our result are promising for future investigations. This should clarify that the scaling we report is not immediately generalizable to more complex problems, of which we will investigate in future reports.

Final Comment:

From above reasons—and since the work is still technically sound and interesting—I believe this manuscript is better to be published in a specialized journal. There, the community will more likely appreciate solid but incremental progress on fiber-based nonlinear photonic system or Ising machine field.

Reviewer #1

(The full report of this Reviewer is quoted in blue below.)

This Reviewer reviewed our original submission to Nature Photonics. The comments below are in response to the revisions we made to transfer the manuscript to Nature Communications. In the first paragraph of their report, copied below, they make clear that we have satisfactorily addressed their concerns.

I reviewed the revised manuscript by Quinn et al., “Coherent Ising Machine Based on Polarization Symmetry Breaking in a Driven Kerr Resonator.” In my original report, I asked the authors to (i) quantify the impact of a non-perfect π defect, (ii) discuss the threshold for the onset of the polarization symmetry-breaking regime relative to other Kerr-driven phenomena, and (iii) expand their outlook on scaling and possible all-optical coupling architectures. All these points have been addressed in the revised manuscript. The tolerance to phase-defect imperfections is now explicitly discussed and supported by references [29, 32]; the power threshold is compared with that of modulation-instability and soliton formation; and the Discussion section now includes a perspective on future all-optical coupling schemes.

This Reviewer then comments on the original report of Reviewer #3, who has not re-reviewed our transferred manuscript. Again, the changes we have implemented are found to address the concerns that were raised by Reviewer #3.

I also find Reviewer #3’s criticism valuable and constructive. The authors have undertaken a consistent and careful revision in response, and I find that the new version substantially improves the manuscript. In particular, the updated statistical analysis (Fig. 4) and the demonstration of continuous one-hour operation without post-selection highlight a clear enhancement and a genuine focus on system robustness.

Above, we note in particular that Reviewer #1 considers our work to be a “clear enhancement” with respect to prior art. We thank them for these nice comments.

The work’s novelty lies in the realization of a symmetry-protected, polarization-based Ising machine, introducing a fundamentally new spin-encoding mechanism distinct from previous phase-based CIMs. The added discussion on π -defect tolerance and threshold comparison further consolidates the conceptual foundation.

While the overall computational performance and benchmarking remain below the state of the art, such optimization is primarily a technical goal. The key contribution of this paper is the demonstration of a viable and conceptually new mechanism for topologically protected spin encoding. The manuscript now provides a well-validated proof-of-concept of a symmetry-protected optical computing paradigm, which falls well within Nature Communications’ remit to publish technically sound and conceptually significant advances.

In this final remark, Reviewer #1 confirms the suitability of our manuscript for Nature Communications. Overall, all of Reviewer #1’s comments are positive and no further questions or concerns are raised. For this re-submission, we have therefore not made any further changes to the manuscript in relation to Reviewer #1.

Reviewer #4

(The full report of this Reviewer is quoted in blue below.)

The manuscript is about implementing an Ising machine based on polarization symmetry breaking in a fiber cavity, and demonstrating the ability of this system to solve optimization problems by minimizing the Ising energy.

The paper has already been reviewed (for Nature Photonics) by two reviewers, one of which is rather positive while the other reviewer has some questions about novelty and about the appropriateness of the benchmark problems used in the paper.

The topic of the paper, being a combination of photonics, non-linear dynamics and computing, certainly fits within the scope of the journal Nature Communications. The paper is well-written and clearly structured and the response to the reviewers' questions address most of their concerns. I believe the paper to be of interest to the research communities working on Ising machines and those working in the field of non-linear optics. The main innovation of the paper for me is that the suggested approach leads to a much-enhanced stability, even in the case of using non-ideal components and/or when drift is present. Moreover, using topology to avoid asymmetries can be an interesting scheme for other Ising machine implementations.

We thank the reviewer for these positive comments. We appreciate the recognition of the clarity of our communication, the acknowledgment that the paper would be of interest to the Ising machine and nonlinear optics communities, and that it fits within the scope of Nature Communications. In their report, the Reviewer also clearly points out the major highlight of our work, which is the novelty of the approach in enhancing stability and mitigating asymmetries for Ising machines.

Although I do agree with most of the claims of the paper, I think some more work is needed to further substantiate the conclusions. My main concern is related to possible unintentional effects influencing the proper operation of the Ising machine. The authors clearly motivate that the localized defect results in bias free spins in the case that there is no coupling between the spins, which is an important effect. But can they also prove/show that this leads to the implementation of the intended Ising Hamiltonian when there is coupling? Related to this issue is the remark made by the authors that the balancing of the asymmetries remains robust even when there is a deviation of the π phase defect. Again, without coupling this statement seems to be justified, but can the authors further elaborate on this issue when operating the coherent Ising machine (i.e. when there is coupling)? Would it then be possible that the Ising machine actually oscillates between 2 ground-states that are both (slightly) different from the ground-state of the initial problem at hand?

The Reviewer raises an insightful point about the dynamics of our system when operating with the Ising perturbations. They are correct in that, while we have done analytical investigations of the protected symmetry breaking regime, these investigations did not account for the coupling between the spins, and associated perturbations to the driving pulses, inherent to solving the Ising problem. As such, it is fair to ask whether the full system, with the π phase shift topological defect and the non-idealities (biases and drifts), models the expected Ising Hamiltonian in presence of the coupling. To address this, we have extended the derivation of the symmetry-protected mean-field model of our resonator (as was derived in the methods of Ref. [29]) to include the coupling. We find that the only difference is an effective driving term for the E_2 mode, which over two round trips has the form as shown on the right of the equation below (written here for each individual spin i)

$$E_{2,i}^{(m+2)} = E_{2,i}^{(m)} + \dots + ig\sqrt{\theta} E_{\text{in}}(\sin \chi) \left(\sum_j J_{ij} \left[|E_{+,j}^{(m+1)}|^2 - |E_{+,j}^{(m)}|^2 \right] \right). \quad (1)$$

where m is the round-trip index. The key point to observe is that the difference between the two mode intensities that appear in the sum on the right exactly corresponds to the differential intensity I_D which defines the spin σ_j in our system. Hence, the above shows that the perturbation that we apply to spin σ_i has the form $\sum J_{ij}\sigma_j$. In the Hamiltonian picture, this demonstrates that we correctly implement the intended Ising problem, irrespective

of imperfections in the π phase shift defect. We can also note that, as opposite spins (+1 versus -1) correspond to an exchange of the circular polarization modes, $|E_+|^2 \rightleftharpoons |E_-|^2$, they perturb the system in exactly opposite ways and the system is inherently symmetrical and balanced. In that respect, and in reference to the last part of the Reviewer’s comment above, it makes clear that the system does not exhibit two slightly different ground states even in presence of imperfections. We also note that the Ising energy, $\sum J_{ij}\sigma_i\sigma_j$, is invariant when inverting all the spins, $\sigma_k \rightarrow -\sigma_k$.

The above equation and discussion have now been included at the end of the Methods and we refer to it in the Experimental implementation Section. We thank the Reviewer for their insightful comment as this addition highlights an important aspect of our Ising machine.

Some other, minor comments are

1. Related to the comment on the threshold and power dissipation, it would be illustrative to add some information about injected power and power dissipation in the methods section.

Our system is not very sensitive to the exact power used and works across a range of injected powers, which we can vary by a factor of three or more without impacting performance. Also, the relation between average power (which we can easily measure) and peak power (which is what is ultimately important) is complicated by a number of factors, including the number of spins, the exact pulse duration and shape (which can vary day-to-day depending on the exact settings of the EO comb source), as well as some contribution from amplified spontaneous emission (ASE) noise from the amplifier (which is not coupled into the resonator as it is not coherently enhanced). The fraction of ASE also depends on the number of spins, which affects amplifier’s saturation. We have estimated peak driving powers based on simulations, and we find that peak powers ranging from 0.4 to 1 W give good agreement with the experiments. We now quote these values in the Methods.

With regards to power dissipation, our resonator exhibits a finesse of 42, which is an improved value to what was stated (about 40) in the Experimental implementation section, and which corresponds to about 15 % power loss per roundtrip. This is now also mentioned in the Methods. We also provide more details about the role of dissipation on the time-to-solution in our reply to comment 5 below.

2. To make results reproducible, please specify all parameter values used for the simulations in Figs 3, 4 and 5. There is an index missing in $\phi^{(m+1)}$ in Eq (3).

As requested, we have now included all parameters used for both the simulations and experiments in the Methods. Note that Fig. 5 contains only experimental data.

Regarding the missing index in Eqs (2–3), the Reviewer is correct that all intracavity fields appearing in these expressions should have an index (e.g. i) to designate the spins, ie. $E_{1,i}^{(m+1)}$ etc. We had initially omitted these indices to simplify the notation (and instead stated that the equations were *for each spin*). We now explicitly use spin indices across that Section.

3. How do you match the value of parameter g (the coupling strength) in the model with the experimental parameters to match the experimental and the numerical distributions shown in Fig 4 (a) and (b)? Is this correspondence derived from other, independent measurements or are the parameters matched based in these distributions themselves?

The value of the parameter g used in the numerical simulations of Fig. 4 is used as a fitting parameter to match the experimental data. This is necessary because it is difficult to characterize the phase modulation actually applied to the driving field as it depends in a non-trivial way on the applied voltage. We now explicitly state the value we used in the Methods. Other parameters (e.g. resonator finesse, pulse duration, fibre nonlinearity, . . .) are directly measured from the experiment. The Methods have been updated to reflect this.

4. When discussing the use of an FPGA for all-to-all coupling, the coupling will typically be much slower than the resonator's round trip time. Do you foresee extra difficulties with matching the feedback signal to the correct round-trip? Will this require a precise clocking of all signals?

Our current resonator round-trip time is 273 ns, which corresponds to a free-spectral range of 3.66 MHz. Commercial FPGAs can easily provide updates at such a rate and so we do not foresee any issues regarding this aspect. For example, the commercially available XILINX Ultrascale+ FPGA platform can provide up to 1 GHz of bandwidth and so would operate significantly faster than the round-trip time of our resonator, allowing the application of perturbations every resonator round-trip. The process can be further accelerated by calculating the feedback term for each pulse in parallel, which would be highly efficient on a chip like the XCVU9P which has 6,840 DSP slices. As this is not directly relevant to the current results, no changes have been made to the manuscript in regards to this question.

5. Related to the comment on the number of round trips needed to reach a solution, it is true that the system of the paper is fast as the round-trip time is short, leading to an annealing time of only 60 μ s. Still, the fact that a (somewhat) larger number of iterations is needed is interesting in itself, as it seems to suggest that the inherent time-scale at which the Ising machine fluctuates is somewhat slower (as expressed in number of iterations). Can the authors explain these differences and pinpoint which system parameters influence this inherent time-scale?

We thank the reviewer for this insightful observation. They are correct that, under our current operating conditions, the annealing time in our system appears slower (with respect to some prior art), as a greater number of iterations (resonator round-trips) are required for the spin states to reach their final configuration.

The two primary system parameters influencing this convergence rate are the time over which we sweep the driving laser frequency across the SSB bifurcation — also known as the annealing time T_a — and the photon lifetime t_{ph} of the resonator. The latter is given by $t_{ph} = t_R \mathcal{F} / (2\pi)$ where $t_R = 273$ ns is the round-trip time and $\mathcal{F} = 42$ is the finesse of the resonator, giving $t_{ph} = 1.8$ μ s. The photon lifetime determines the time scale over which the intracavity power can vary (the annealing time cannot be smaller than the photon lifetime), and is dependent on power dissipation (the finesse is inversely proportional to the resonator losses). Intuitively, a resonator with a very high finesse is so weakly coupled to the external driving that the intracavity field evolves and approaches steady states very slowly.

In comparison to Ising machines based on degenerate OPOs, our finesse is comparatively large, which has the advantage of requiring less driving power but comparatively slows down the dynamics. However, we can easily speed up the dynamics by increasing the round-trip losses, which results in a lower finesse.

To illustrate this, Figure A (below) presents simulation results for our Ising machine over 200 trials of a 64-spin one-dimensional Ising model with two different resonator finessses (all other parameters being the same, including identical normalized driving power to keep identical SSB characteristics). Figure A(a,b) corresponds to the parameters of our current experimental setup (finesse of 42), while Figure A(c,d) uses a significantly lower finesse of 15. The Ising machine converges much more rapidly in the lower-finesse case, as expected (and also faster than some prior DOPO-based Ising machine). Experimentally, we can readily achieve such a reduction in finesse by introducing controlled bend losses in the fiber system, so operating in this regime is entirely feasible. However, we note that this ‘speed-up’ affects the energy distribution [compare the insets of Fig. A(a) and (c)], with a lower probability of reaching the ground-state in the lower finesse case, which is not necessarily desirable. Clearly there is a trade-off between speed and solution quality. We now comment on this aspect in the Discussion.

Figure A: Comparison of high and low finesse resonators for our symmetry-breaking Ising machine. Lowering the finesse leads to faster convergence, but worse energy distributions.